# Roles of Peripheral Nerves in Tumor Initiation and Progression

**DOI:** 10.3390/ijms26157064

**Published:** 2025-07-22

**Authors:** Claudia Giampietri, Elisa Pizzichini, Francesca Somma, Simonetta Petrungaro, Elena De Santis, Siavash Rahimi, Antonio Facchiano, Cinzia Fabrizi

**Affiliations:** 1Department of Anatomy, Histology, Forensic Medicine and Orthopaedics, Sapienza University of Rome, 00161 Rome, Italy; claudia.giampietri@uniroma1.it (C.G.); elisa.pizzichini@uniroma1.it (E.P.); francesca.somma@uniroma1.it (F.S.); simonetta.petrungaro@uniroma1.it (S.P.); elena.desantis@uniroma1.it (E.D.S.); 2Istituto Dermopatico dell’Immacolata, IDI-IRCCS, 00167 Rome, Italy

**Keywords:** peripheral nerve, nervous system, Schwann cells, melanoma, pancreatic ductal adenocarcinoma, cholangiocarcinoma, hepatocarcinoma

## Abstract

In recent years, a long list of relevant studies has highlighted the engagement of the nervous system in the fine-tuning of tumor development and progression. Several authors have shown that different types of nerve fibres (sympathetic, parasympathetic/vagal or somatosensory fibres) may contribute to tumor innervation affecting cancer initiation, progression and metastasis. A large presence of nerve fibres is frequently observed in tumors with respect to the corresponding healthy tissues. In this regard, it is worth noting that in some cases a reduced innervation may associate with slow tumor growth in a tissue-specific manner. Current studies have begun to shed light over the role played in this specific process by Schwann cells (SCs), the most abundant glial cells of the peripheral nervous system. SCs observed in cancer tissues share strong similarities with repair SCs that appear after nerve injury. A large body of research indicates that SCs may have a role in shaping the microenvironment of tumors by regulating the immune response and influencing their invasiveness. In this review, we summarize data relevant to the role of peripheral innervation in general, and of SCs in particular, in defining the progression of different tumors: melanoma that originate in the skin with mainly sensory innervation; pancreatic and liver-derived tumors (e.g., pancreatic adenocarcinoma and cholangiocarcinoma) with mainly autonomous innervation. We conclude by summarizing data regarding hepatocarcinoma (with anatomical predominance of small autonomic nerve fibres) in which the potential relationship between innervation and tumor progression has been little explored, and largely remains to be defined.

## 1. Introduction

In vertebrates, peripheral nerves are widespread and distributed to the most diverse regions of the body thereby providing a bridge between the central nervous system (CNS) and the peripheral organs. These nerves emerge from the brain through the cranial nerves and from the spinal cord as spinal nerves. The peripheral nervous system (PNS) is typically subdivided into the autonomic nervous system (ANS) and the somatic nervous system consisting of both sensory and motor fibres. The ANS includes the sympathetic nervous system (SNS), parasympathetic nervous system (PSNS), and enteric nervous system (ENS). The ENS forms a reticular structure capable of functioning independently of the rest of the nervous system and is primarily responsible for regulating digestive processes (Figure 1).

Each peripheral nerve frequently contains both motor and sensory axons of various calibres. These axons are enveloped by the most abundant glial cells in the PNS, the Schwann cells (SCs). The myelinating SCs repeatedly wrap around the large diameter axons forming the myelin sheath. Between each SC and adjacent ones, the myelin sheath is interrupted by bare regions of the axon known as nodes of Ranvier. Non-myelinating SCs include Remak SCs that ensheathe multiple small-calibre axons and terminal SCs that chemically and physically support neuromuscular junctions. Both myelinating and non-myelinating SCs play a central role in governing axonal regeneration after injury.

As already anticipated by Rudolph Virchow in the 19th century, tumor progression exhibits extensive parallelism with the development of inflammatory and regenerative processes [1]. This initial concept was greatly expanded in later years to the point of considering tumors as wounds that do not heal [2]. Thus, tumor development has many points in common with the normal processes that direct and guide the regeneration of organs and tissues that have been wounded. In this respect, numerous reports linking innervation with tumor progression have been accumulating in recent years. Notably, many of these recent data focus on the role played in this process by SCs.

## 2. Schwann Cells: Master Architect of Peripheral Nerve Regeneration and Beyond

SCs originate from migratory neural crest cells (NCCs) in a multistep process involving several intermediate stages. The NCCs form a population of multipotent stem cells of ectodermal origin which generate an extraordinary variety of cell types that in addition to SCs comprises other glial cells such as enteric glia and ganglionic satellite glia. In addition, different populations of neurons of the PNS are originated from NCCs, including sensory neurons, postganglionic sympathetic and parasympathetic neurons, and enteric neurons. Melanocytes, some endocrine derivatives and mesenchymal cell types are also descended from NCCs [3].

Beginning at embryonic day (E) 12.5, murine NCCs start giving rise to Schwann cell precursors (SCPs) [4,5,6,7]. SCPs lack a basal lamina, are located proximal to the growing nerve tip, and contribute in guiding axons towards their targets [5,6]. At later stages of development (from E14.5 onwards), SCPs give rise to immature Schwann cells (iSCs), a cell type that is detected up to birth [5]. Shortly before birth, through a process called “radial sorting” [8] iSCs that contact large diameter axons, a source of high levels of neurogulin 1 (NRG1), become pro-myelinating and then myelinating SCs. Conversely, iSCs that contact small-calibre axons, which produce low levels of NRG1, form the non-myelinating (Remak) SCs [9,10]. Although data on the embryonic origin of SCs in humans are more limited, they nevertheless suggest a high degree of concordance with those obtained in mice. Nerve repair SCs are non-myelinating SCs that derive from myelinating and Remak SCs at sites of nerve damage. Following axonal injury, SCs undergo a process of de-differentiation that retraces backward some stages of embryonic development resulting in the formation of an SC repair phenotype that is nonetheless distinct from embryonic progenitors. Repair SCs proliferate and migrate to form longitudinal alignments of cells (named bands of Büngner), providing essential guidance for regenerating axons [11,12].

## 3. Innervation and Tumor Progression

The link between innervation and tumor progression has been described and discussed in a large number of articles [13,14,15,16,17,18,19], and key data have been collected in numerous reviews, including recent ones [20,21,22,23]. Different studies show that solid tumors can induce neurite extensions and attract peripheral nerves. These processes, known as neurogenesis or axonogenesis, lead to aggressive tumor characteristics and are usually associated with poor clinical outcomes [24,25,26,27]. However, data currently available are not entirely consistent probably due to the variety of tumors, their stage, sampling, and investigation methods. The positive association between nerve fibre density and tumor recurrence risk has been established in numerous cancers arising from different organs (prostate, stomach, colon–rectum, head and neck) [21]. However, other studies have identified an inverse correlation, identifying certain categories of nerve fibres (sensitive and vagal fibres) as negative regulators of tumor growth (e.g., in melanoma and pancreatic ductal adenocarcinoma, PDAC) [28,29]. The difficulty in quantifying fibres by immunohistochemistry may, at least partly, explain the discrepancy in reported cancer innervation.

The present review attempts to interpret existing data by comparing tumors arising from organs and tissues with a different underlying innervation (i.e., melanoma, pancreatic adenocarcinoma, cholangiocarcinoma and hepatocarcinoma).

The autonomic nervous system (ANS) in general and the SNS in particular seem to act as important elements in the regulation of tumorigenesis. Virtually all organ systems in humans are influenced by the SNS via the catecholamine neurotransmitters, either through the release of epinephrine (adrenaline) and norepinephrine (noradrenaline) by tissue-localised nerve terminals or through the vascular distribution of epinephrine (adrenaline) secreted by the adrenal gland. A large body of epidemiological and experimental data has demonstrated that the SNS-induced stress response plays an important role in the early stages of tumor progression, both through direct effects on malignant cells and indirectly by helping to create a tumor-promoting microenvironment [30,31,32].

Conversely, unilateral vagotomy and vagal sensory fibres inactivation through perineural capsaicin treatment increased metastasis of breast carcinoma without altering the growth rate of the primary tumor, thus indicating a protective role of vagal sensory fibres against cancer [33,34]. In accordance with these data, in PDAC models, cholinergic signalling through muscarinic receptors suppresses pancreatic tumorigenesis [28]. On the other hand, abrogation of the cholinergic input by vagotomy or chemical denervation inhibits the growth of gastric cancer [15].

As with the ANS, the correlation between sensory innervation and tumor growth and progression remains to be defined in many respects. Most work on the link between sensory fibres and cancer has looked at cancer-related pain syndrome and perineural invasion. At present, the factors responsible for cancer-related pain are poorly understood; however, tumor-induced pathological sprouting of sensory nerve fibres and SC abnormalities have been suggested as possible causes [35,36,37,38,39].

As for the perineural invasion or perineural spread, i.e., the spread of cancer cells in and along nerve bundles, the protumoral activity of SCs has been observed in some cases. Contrary to initial expectations, recent studies have demonstrated that SCs play an active role in guiding malignant cells towards nerves and in facilitating perineural invasion. Cocultures of cancer cells and dorsal root ganglia (DRG) clearly show that SCs are responsible for guiding the migration of malignant cells towards nerve fibres [40,41,42,43].

Also, sensory fibres appear to be linked to the pathogenesis of cancer. Remarkably, ablation of sensory neurons of the spinal ganglia has been demonstrated to slow tumorigenesis and to lead to overall survival in murine models of basal cell carcinoma and PDAC [35,44]. Conversely, the growth and angiogenesis of melanoma are accelerated when the activity of sensory neurons is inhibited and an increased expression of sensory neurons-related genes within human melanomas is associated with improved survival [29]. More evidence is needed to understand how sensory neurons may foster or limit tumor growth and spread.

In this respect, it is noteworthy that SC-dependent tumor perineural invasion and tumor-related pain are often associated with a worse prognosis [45,46] indicating pain not only as an alert symptom, but also as an independent prognosis factor, at least in head and neck squamous cell carcinoma [47] and in advanced prostate cancer [48].

To further reinforce the role that pain may play in tumor biology, pain relief drugs such as propranolol (a beta-blocker inhibitor of adrenergic signalling) and botulinum toxin, relieve cancer-related pain and impair tumor growth [49,50].

## 4. Schwann Cells and Cancer

In cancer, recent studies have highlighted an active role of SCs in promoting cancer progression and cancer cell invasion [51,52,53,54,55].

SCs detected in cancer tissues share strong similarities with nerve repair SCs. Non-myelinating nerve repair SCs are characterized by the re-expression of the transcription factor c-Jun, followed by the upregulation of markers such as neurotrophin receptor p75 NTR (NGFR), glial fibrillary acidic protein (GFAP), growth associated protein 43 (GAP43), and Sox2 [42,51,53,56,57]. Tumor-associated SCs have been identified in tumor samples, with their abundance correlating with disease prognosis. Tumor-associated SCs can be reprogrammed by tumor-derived factors to acquire the repair-like phenotype, enabling them to express and release cytokines, chemokines, growth factors, neurotrophic factors, matrix metalloproteinases, and exosomes. These secreted elements and other factors contribute to tumor microenvironment modulation through three main mechanisms: (i) attracting and polarizing immune cells toward an immunosuppressive phenotype, recruiting Myeloid-Derived Suppressor Cells (MDSCs) in the tumor microenvironment and exacerbating the malignant features of cancer-associated fibroblasts (CAFs); (ii) remodeling the extracellular matrix (ECM); (iii) directly influencing the functional activity of malignant cells [40,58,59,60,61].

Furthermore, SC-derived chemokines have been shown to enhance epithelial-mesenchymal transition (EMT) in cancer cells, thereby fostering their migratory potential [62,63,64].

MicroRNAs from SCs exosomes further promote the proliferation, motility, and invasiveness of cancer cells by targeting and blocking specific mRNA within malignant cells [63,65,66,67].

Collectively, tumor-associated SCs have been shown to facilitate tumor growth in vivo and contribute to the establishment of distant metastases (Figure 2).

### 4.1. Melanoma and Innervation: The Role of Schwann Cells

The skin is innervated by numerous afferent fibres and overall constitutes the largest sensory organ of the human body. Skin maintenance and healing depend on proper cutaneous innervation and nerve fibres localized in the skin play a deep regulatory control. The lack of this control, occurring under pathological conditions such as spinal cord injury or diabetic neuropathy, leads to ulcers and delayed wound healing [68,69,70,71,72].

In human skin, non-myelinating SCs are arranged in a complex and dense network located just below the dermo-epidermal junction (200–300 cells/mm^2^) [73,74]. A specific interplay of melanoma and the nervous system is expected, given the embryological origin of melanocytes that may derive either directly from neural crests or, alternatively, from neural crest-derived SC precursors [75,76]. Primary melanoma and primary melanocytic tumors in CNS are a rare occurrence [77,78,79], while brain localization of melanoma metastases occurs frequently. In fact, up to 60% of advanced melanoma may develop brain metastases [80]. Additionally, uveal melanoma is the most frequent intraocular tumor, affecting uvea i.e., the highly innervated vascular middle layer of the eye [81]. This may suggest permissive action of nerve fibres on melanoma growth. On the other hand, genetic ablation or chemical denervation of sensory nerves in mice accelerates melanoma growth in vivo indicating that sensory fibres counteract melanoma progression [39]. Consistently, inhibition by a chemogenic approach of the activity of sensory neurons promotes melanoma growth and intra-tumoral angiogenesis while their excitation induces melanoma regression [29].

However, recent evidence reveals that high density of intratumor nerves parallels with poor prognosis in melanoma patients [82,83]. In this regard, it is worth noting that interaction of melanoma cells with nociceptive neurons promotes axonogenesis leading to tumor innervation which in turn lowers the immune response to the tumor [84].

Sensory ablation decreased lymphoid and myeloid immunosuppressive cells and promoted T-effector cell activation within the tumor microenvironment. The modulation of the immune system appears as the primary mechanism by which sensory neurons support melanoma growth. Remarkably, CD8a-depletion prevents the denervation-dependent antitumor effect, leading to reduced populations of T cells (CD8+, CD4+, Treg) [19]. Under such conditions, melanoma may grow exploiting the immune privilege of the nervous system to evade immune surveillance [85]. The interplay between nerves and the immune system in melanoma is also further supported by data showing that melanoma interacts with nociceptor neurons leading to an increased release of the Calcitonin Gene-Related Peptide (CGRP) which in turn reduces the cytotoxic activity of CD8+ T cells [84], and by data indicating the onset of a neuron-dependent immunosuppressive environment [41].

Within the regulatory action of the nervous system on melanoma, SCs have been shown to play a key tumor-promoting role; in fact, similarly to what occurs within a healing wound, SCs exert their nerve-repair and wound-healing action also within the tumor microenvironment, as a response to the nerve injury reaction activated by melanoma cells. The melanoma-induced SCs activation leads to melanoma growth promotion, indicating SCs as key melanoma growth promoters and immune-suppressive players, thus representing valuable therapeutic targets [53,86]. Another interesting potential therapeutic target is the Glycoprotein Nonmetastatic Melanoma Protein B (GPNMB), which, in addition to its immunosuppressive activity, also has the ability to foster peripheral nerve regeneration. It is overexpressed in melanoma and acts on Erk1/2 and Akt pathways in SCs promoting their proliferation and migration [87].

An additional aspect to consider is that SC precursors act as an ontogenic source for a variety of cell types (such as fibroblasts, melanocytes, neurons, parasympathetic ganglia, the SCs themselves) and that a variety of tumors arise from SCs, i.e., malignant peripheral nerve sheath tumors, schwannomas, neurofibromas and the Devil Facial Tumor Disease (DFTD) [40]. Moreover, the rare occurring intracranial melanotic schwannoma and the rare malignant melanotic nerve sheath tumor (MMNST) require an accurate differential diagnosis from malignant melanoma, indicating the occurrence of overlapping histological and clinical characteristics of melanoma with SC-derived tumors [88,89].

As a final observation, sensory symptoms such as pain and itch, often mediated by SCs, may represent symptoms in non-melanoma [90] as well as melanoma skin cancers [91].

#### 4.1.1. Histopathological Characteristics of Melanoma–Nerves Interplay

Histopathological and immunohistochemical analyses can reveal characteristics of interaction between melanoma and the PNS which potentially could affect the prognosis and therapeutic strategies. The cells of cutaneous melanoma could interact with nerves in different ways reported below.

##### Perineural Invasion

Perineural invasion describes the ability of tumor cells to infiltrate a nerve by passing through the layers of its sheath. Doing so, tumor cells find a favourable environment to travel around the body, contributing to the progression of the disease. Melanoma cells could invade the perineurium showing small nests architecture or as single cells with epithelioid or spindle morphology. Although perineural spread of malignant melanoma is rare, it is associated with a poor prognosis such as an increased rate of local relapse and/or higher probability of lymph nodes or distant metastasis [53,92,93].

##### Neuropathic Changes

Melanoma in an advanced stage causes degeneration of nerve fibres, loss of the myelin integrity and fibrosis that contribute to weakness, numbness and pain. Melanoma cells have been shown to drive the release of macrophage colony-stimulating factor (M-CSF), which elicits resident intraneural macrophages expansion. This macrophage subpopulation seems to be involved in mechanical/cold hypersensitivity and spontaneous nociception [94].

##### Neural Remodeling

Melanoma cells can remodel their own microenvironment changing the structure and density of nerves. The new nerves are recognizable histologically as elongated fibres alongside lesional melanoma cells. This has been demonstrated through immunohistochemical analyses with markers for nerve fibres such as neurofilaments, S100 protein and protein gene product 9.5 (PGP 9.5) [95,96]. Metastasizing and low metastatic melanomas display a different pattern of neural activity in animal models [97].

##### Neurogenic Inflammation

The nerve fibres in melanoma can release pro-inflammatory chemotactic factors that attract inflammatory cells including lymphocytes, mast cells and macrophages that can be detected by histological examination. This process triggers a neurogenic inflammation which results in neuropeptide release and rapid plasma extravasation and oedema. The clinical implication of neurogenic inflammation is tumor progression and can also cause immune evasion, which can be exploited for target therapy [40,98,99].

##### Neurotrophic Factors and Receptors Expression

Melanoma cells can express neurotrophic factors, such as the nerve growth factor (NGF) and brain-derived neurotrophic factor (BDNF), and their corresponding Tyrosine kinase receptors (TrkA and TrkB) that can be detected by immunohistochemistry. The neurotrophic factors and their receptors are associated with the progression of melanoma and can be used for new therapeutic strategies. In brain-metastatic melanoma cells, NGF has been shown to promote invasion by increasing the production of extracellular matrix-degrading enzymes, suggesting that tissues rich in NGF and other neurotrophins, such as the brain, support melanoma invasion and survival through a potent chemotactic activity. Melanoma exploits the immune privilege inherent in its developmental origin and induces immunosuppression impairing the formation of the tumor-specific T cell memory. The NGF drives immune cell exclusion in the melanoma tumor microenvironment. Inhibiting the NGF renders melanomas susceptible to immune checkpoint blockade therapy and fosters long-term immunity by activating memory T cells with low affinity [85,100].

### 4.2. Innervation in Liver and Pancreas

Nerve fibres in the liver and pancreas belong to the visceral autonomic sympathetic nervous system (SNS) and parasympathetic nervous system (PSNS). Innervation by SNS occurs via the splanchnic nerves, and innervation by the PSNS occurs via the vagus nerve. Both nerve divisions contribute efferent (motor) fibres (for instance, to the walls of the blood vessels, the pancreatic ducts, and the pancreatic acini) and visceral afferent (such as for pain) fibres. In the liver, the afferent fibres provide the CNS with information about the metabolite composition of the portal vein. On the other hand, the efferent fibres control the metabolism, blood flow and bile secretion [101,102]; to note, autophagic processes are also key actors in regulating liver metabolism [103]. The biliary tree is highly innervated via the ANS: the SNS activates and increases the proliferation of hepatic stellate cells (HSCs) while PSNS cholinergic fibres regulate cholangiocyte secretion and proliferation. In fact, α-adrenoblockers have been shown to be promising among strategies able to reduce HSC activation and fibrosis [104,105] and inhibitors of the SNS are potential candidates for the treatment of cirrhosis [106,107].

#### 4.2.1. Pancreatobiliary Tract Cancer: The Role of Nerves Fibres and Schwann Cells

##### Perineural Invasion

Both PDAC and cholangiocarcinoma (CCA) are aggressive cancers, and they share many biological features including neurotropism, which results in frequent perineural invasion [108]. It has been reported that almost all PDAC lesions, as well as about 75% of CCA, undergo perineural invasion. Both in PDAC and CCA, perineural invasion is associated with a worse prognosis and shorter survival. SCs guide the migration of cancer cells towards nerves and have been shown to promote perineural invasion in PDAC [62,108,109,110].

##### Neuropathic Changes

SCs not only play a key role in cancer progression, but also in cancer-related pain. In PDAC, in the early stages of cancer, interleukin-6 (IL-6) secreted by SCs was linked to analgesia since activated SCs exhibited a transcriptomic profile with anti-inflammatory and anti-nociceptive features. IL-6 blockades inhibited SCs activation. The initial activation of SCs therefore suppresses the activity of microglia and astrocytes within the spinal cord, thus inhibiting afferent pain fibres. In PDAC, patients with more SCs experience less pain and the transient initial state of analgesia may result in a possible diagnostic delay [111,112,113]. Conversely, as the tumor progresses, SCs secrete several pain-related molecules, such as the NGF, that promote the activation of primary sensory neurons, thus increasing nociceptive activity [37,38,48,114].

##### Neural Remodeling

Nerve fibres are present in the tumor microenvironment in both PDAC and CCA. Their presence, together with that of different nonmalignant cells such as fibroblasts, immune cells, and blood or lymphatic vessels, plays an important role in carcinogenesis. For instance, SCs with a repair phenotype may aid tumor growth through a direct interaction with cancer cells and ease the onset of an immunosuppressive microenvironment. In vitro SCs induce an M2-phenotype in macrophages, thus possibly leading to worse clinical prognosis in malignancies [20,115,116,117,118,119]. In PDAC, neurogenesis is a novel studied biological phenomenon since cancer cells may induce nerve growth and innervation through multiple mechanisms, including secretion of neurotrophic factors such as NGF, BDNF, and the glial cell-derived neurotrophic factor (GDNF). In PDAC, cancer-related neurogenesis via the “proangiogenic vessel guidance factor placental growth factor” has been recently demonstrated; this promotes neurite outgrowth and attracts tumor cells towards nerves. In addition, it has been proposed that cancer cells might recruit neural progenitors to facilitate their maturation into adrenergic infiltrating nerves which in turn may stimulate angiogenesis. Interestingly, the cancer stem cell pool seems to be expanded via intratumoral PSNS nerves which can induce Wnt-β-catenin signals [120,121,122].

It is not known whether cancer-related neurogenesis also occurs in CCA. Remarkably, in CCA small nerve fibre density may be considered a novel prognostic biomarker. Nerve fibre density is different from perineural invasion since it describes the density of small nerve fibres without cancer invasion. It has been reported both in perihilar CCA and in intrahepatic CCA patients that high nerve fibre density is associated with better patient survival and high CD8+PD-1+ expression within the tumor microenvironment. This underlines how the cellular and molecular immune and nervous constituents may synergize within the tumor microenvironment [123]. In PDAC, a high density of PSNS nerve fibres correlates with tumor budding and poor survival rates [124], while in a metastatic PDAC mouse model, vagotomy fosters accelerated tumorigenesis [28]. Conversely, ablation of sensory innervation has been shown to be effective to slow initiation in the early stages of cancer [108] making the role of nerve fibres in tumor progression particularly complex to interpret.

Regarding the role of SCs in PDAC, they have been shown to actively migrate toward cancer cells. The presence of SCs has been demonstrated in precancerous PDAC tissues. More in detail, SCs secrete proteins, such as matrix metalloproteinase 2 (MMP-2), cathepsin D, plasminogen activator inhibitor-1 or tansforming growth factor (TGF), thus promoting pancreatic cancer cell aggressive properties, proliferation and invasion. SCs can contact cancer cells, and recruit macrophages that degrade the perineurium promoting perineural invasion [37,112]. Regarding the role of SCs in CCA, we recently demonstrated that SCs assist in promoting cancer aggressiveness. In fact, we observed increased migration/invasion and survival/proliferation of CCA cell lines exposed to the secreted factors of SC primary cultures. We also found EMT as well as upregulation of key oncogenes and downregulation of tumor suppressors in CCA cells after co-culture with SCs. It emerged that many of the regulated proteins are under the control of TGFβ and that a TGFβ receptor antagonist is able to block the Schwann-cell-induced migration/invasion of CCA cells [62].

##### Neurotrophic Factors and Receptors Expression

SCs make up approximately 90% of the endoneural space in peripheral nerves. They produce a variety of growth factors such as the NGF, BDNF and GDNF. The NGF is also broadly expressed by tumor cells, inflammatory cells and immune cells. Activation of NGF/TrkA signaling therefore induces tumor progression and either the NGF or TrkA may be a therapeutic target against PDAC [37].

#### 4.2.2. HCC and Innervation

##### Perineural Invasion and Neural Remodeling

The role of the PNS in hepatocellular carcinoma (HCC) remains unclear and poorly investigated. The cause may stem from HCC low neurotropic tendencies, with respect to PDAC and other gastrointestinal cancers, or to its uncommon use of perineural invasion as a primary metastatic route [82,92]. However, it is worth noting that perineural invasion might be just one of the many phenomena in which the PNS is involved in tumor progression. Low neurotropic tendencies can partly be attributed to the anatomical predominance of small ANS fibres over major nerve branches within the liver parenchyma [102,125,126]. Ueda and colleagues were the first to describe neural invasion in the portal tract of the liver in an HCC patient [126]. Subsequently, Kanda et al. identified dense S-100 and synaptophysin positive nerve fibres in the capsule of HCC [127]. Remarkably, in 2024, Mandal and colleagues revealed a large abundance of intratumoral nerves in an orthotopic HCC mouse model [128].

Regarding the prognostic implications of both perineural invasion and nerve fibre density in HCC patients, recent literature presents uneven information, likely due to differences in patient cohorts, selected parameters and methodologies, as well as divergent operational definitions of neural involvement within the tumor microenvironment [129]. In the study of Liebl et al., perineural invasion was reported among 6% of HCC subjects, where its sole presence did not show any prognostic power. However, HCC patients with an increased neural invasion severity, defined with a deeper cancer cells invasion into nerve connective tissue, exhibited significantly worse survival [129]. Concerning smaller nerves not invaded by cancer cells, the group of Bednarsch et al. explored the impact of nerve fibre density on oncological survival in a cohort of HCC patients undergoing liver resection for curative intent; unlike other types of cancer (i.e., gastric and colorectal adenocarcinomas, PDAC) in which the presence and density of nerve fibres in the tumor microenvironment is known to have an important prognostic value, nerve fibre density did not predict long-term outcomes in HCC patients [130]. Contrasting data comes from the study of Zhang et al., in which the authors found that HCC, with respect to non-cancerous liver tissues, presented higher levels of tyrosine hydroxylase (TH), correlated with worse clinical features, thus suggesting a link between ANS fibre density and poorer prognosis [131]. According to Wang et al., perineural invasion could partially explain HCC spread to bone tissues, since a significantly higher nerve density was found in HCC bone metastases compared with primary tumor tissues [132]. However, the idea of a link between peripheral nerves and bone metastases is mainly based on histopathological observations, so a functional study is needed for confirmation.

##### PNS and Liver Chronic Inflammation

Liver cancer often arises in the context of chronic inflammation and fibrosis [133,134]. It therefore seems crucial to understand the direct connection between the hepatic nervous system and processes paving the way for hepatocarcinogenesis. SNS hyperactivity relates to liver damage [135] and fibrogenesis [107], primarily acting on HPC [136], and promotes an inflammatory milieu that supports tumor growth. Noteworthy, in an induced-HCC rat model, SNS denervation decreased both fibrosis and tumor formation [137]. On the other hand, PSNS neurons protect from hepatic damage and inflammation [138] and have been proven to support liver regeneration [139]. Surgical resection still stands as one of the primary curative treatments for HCC, yet its recurrence rates reach up to 70% in patients following surgery [140]. Therefore, future studies must investigate the role that the ANS might play in post-operative repair processes.

##### PNS and Tumor Growth

Cholinergic receptors are highly expressed in HCC cells, where they foster proliferation, EMT, invasion, and resistance to apoptosis, pointing out a potential therapeutic target [131,141]. On the other hand, acetylcholinesterase, the enzyme responsible for acetylcholine inactivation, has been characterized for its role as a tumor suppressor, providing novel therapeutic insights [142]. Numerous studies have documented the important role of adrenergic transmission in the pathogenesis of HCC. The expression of β2- Adrenergic Receptors (AR) is significantly upregulated in HCC liver tumor tissues and cell lines, and it is highly associated with poor prognosis in patients [131,143]. β2-AR have been shown to sustain HCC cells proliferation, survival and chemoresistance [144]; hence, the use of β-AR blockers or sympathetic denervation represents a promising therapeutic strategy [137,145,146]. Additionally, non-tumor tissue from HCC livers showed an increased density of α1-AR compared to healthy livers, suggesting that adrenergic transmission might predispose hepatocytes to aberrant proliferation [143].

The hyperactivation of the SNS is furthermore responsible for hepatocarcinogenesis in the context of chronic inflammation, as the stimulation of α1-AR on Kupffer cells promotes IL-6 and TGFβ production and maintains a pro-inflammatory microenvironment favourable for tumor development [137]. Additionally, HSCs in HCC tissues present elevated levels of β2-AR, which contribute to their activation and, consequently, to the malignancy of cancer cells [147]. A different perspective comes from the work of Liu et al., demonstrating that the activation of SNS neurons and β-AR protected from liver cancer mice kept in stressful environments, via reducing inflammation and enhancing anti-tumor immunity [148].

##### Neurotrophic Factors and Receptors Expression

A recent study published by Wang et al., notably pointed out eight key neurotransmitter-related genes strongly linked to HCC pathogenesis. Based on this finding, the authors provided a prognostic model that accurately forecasts the severity of HCC. Their layout revealed significant alterations in processes involved in cellular metabolism and immune response in high-risk HCC cases [149]. Furthermore, the group of Zhang et al., 2022, categorized HCC into two subtypes based on the expression of neural-related genes, which vary in terms of prognosis, clinical stage, immune regulation and critical signalling pathways [150] emphasizing the critical crosstalk between the nervous system and cancer in HCC.

Besides neurotransmitters, several neurotrophic factors have been recently recognized as contributors to HCC, with reference to the BDNF and NGF. The BDNF and its receptor TrkB have been found to be specifically expressed in HCC cell lines and tumor tissues and to induce neovascularization and cancer cell survival/invasion [151,152]. Kishibe et al. found the NGF to be highly expressed in focal hepatocytic lesions from early stages of carcinogenesis and regeneration, but not in adult and developing livers. Moreover, NGF TrkA receptors and nerve fibres were particularly abundant in the walls of tumor-associated arteries, indicating a possible role for the NGF in angiogenesis [153]. The NGF is reported to be elevated in tumor cells of HCC, while both types of its receptors (TrkA and p75NTR) are present in Kupffer cells, endothelial cells and hepatic stellate cells, only in the liver cancer tissue [154,155]. According to Yuanlong et al., however, the expression of p75NTR decreased significantly in HCC tissues, as compared with their adjacent non-cancerous counterparts, and in various human HCC cell lines, making this receptor a potential candidate as a tumor suppressor [156]. Given controversial evidence, it is crucial to carry out more detailed studies on the role of different neurotransmitters in HCC progression; it would also be intriguing to explore the role of glial cells and SCs, as it remains entirely unexplored.

## 5. Conclusions

The role of PNS in cancer has not been completely clarified, making its impact on tumor biology and patient outcomes across common cancer types not fully understood (Table 1).

It is conceivable and plausible that a direct relationship exists between the neurotrophic capabilities of SCs in fostering regeneration after injury and their ability to aid tumorigenesis. The molecular mechanisms activated in the two other events could be very similar (Figure 3). Therefore, SCs traditionally known for their support of axonal regeneration after injury are now also being considered for their potential involvement in tumor progression.

This could lead to new therapeutic strategies that target the interactions between SCs and the tumor microenvironment, as well as novel diagnostic and prognostic approaches assessing nerve density. It is interesting to note that perineural invasion is a common finding in melanoma (especially desmoplastic subtypes), PDAC and CCA, being a marker of aggressiveness and poor prognosis, while in HCC it is a rare and less relevant finding. The role of PNS in HCC remains unclear and poorly investigated and further research is needed to understand possible involvement of SCs in HCC.

While the gene and protein profiling of SCs derived from malignant peripheral nerve sheath tumors as compared to normal SCs has been reported [157], the genomic and proteomic analysis of these glial cells within the context of different tumors remains to be investigated. It would be of interest to determine whether SCs occurring in solid tumors share common features or are instead related to the tumor type and stage. In this context, as already mentioned, SCs appear to have an analgesic effect in the early stages of tumor growth, while increasing nociceptive activity in the later stages. The specific characteristics of SCs present in the early/late stages of tumor progression remain to be determined.

It is notable that recent research has convincingly demonstrated the SCs’ active involvement in modulating immune responses in a variety of pathological conditions, including neuropathic pain as well as cancer. Chemokines and cytokines secreted by SCs attract and influence immune cells. Specifically, SCs produce signals that attract M2-type macrophages (which suppress immune activation) and recruit MDSCs, thereby enhancing their capacity to inhibit T-cell proliferation. SCs can also directly suppress effector T cells, which in turn can push them towards a regulatory or exhausted state [158]. The immunosuppressive activity of SCs certainly constitutes a potential new therapeutic target to be explored in future studies.

## Figures and Tables

**Figure 1 ijms-26-07064-f001:**
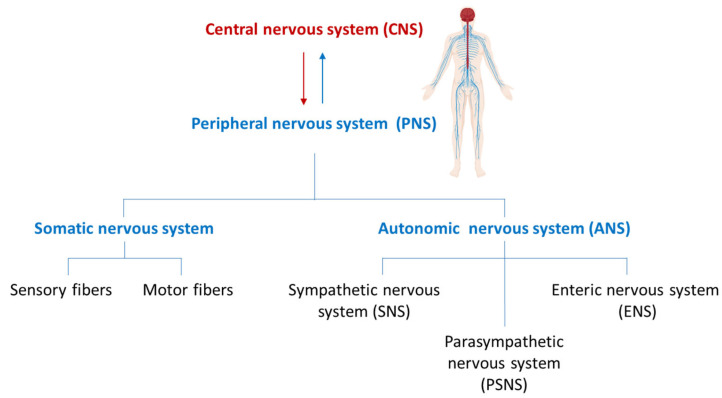
Structural organization of the nervous system.

**Figure 2 ijms-26-07064-f002:**
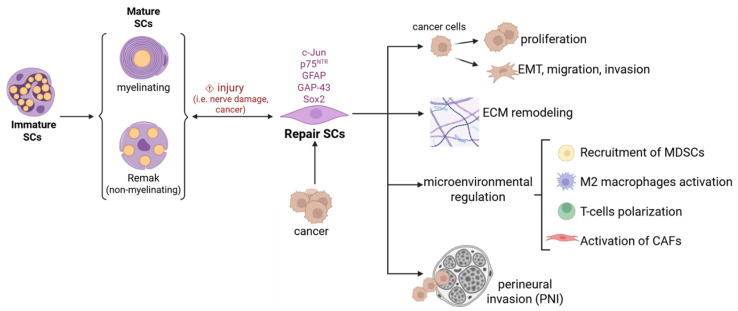
Schwann cell plasticity in cancer pathogenesis. Representation of the main mechanisms through which Schwann cells, reprogrammed to a repair phenotype by cancer, contribute to its progression by acting directly on cancer cells or indirectly on the tumor microenvironment.

**Figure 3 ijms-26-07064-f003:**
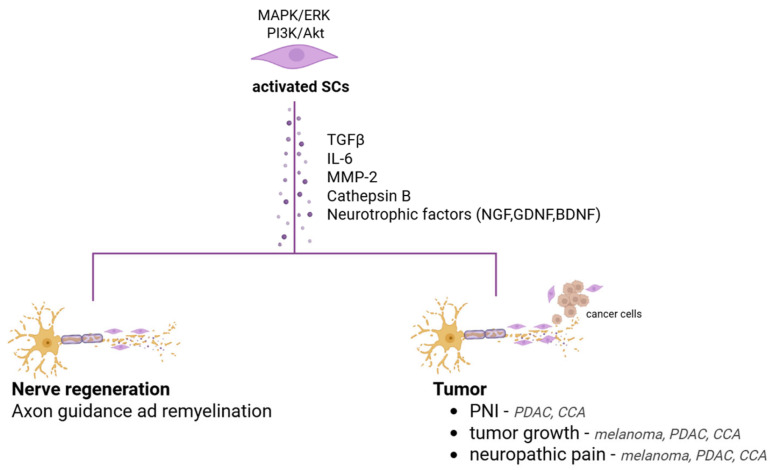
Key molecular pathways and secreted factors of activated SCs that potentially contribute to both nerve regeneration and tumor progression in PDAC, CCA and melanoma.

**Table 1 ijms-26-07064-t001:** Summary of studies regarding the role of innervation in different types of tumors (melanoma, PDAC, CCA, HCC). The pro-tumoral role of Schwann cells, relevant neurotrophic factors regulating tumor innervation and possible therapeutic targets are indicated for the different tumors.

	Pro-Tumural Role of SCs	Sympathetic Fibres	Parasympathetic Fibres	Sensory Fibres	Relevant Neurotrophic Molecules	Potential Drugs	References
**Melanoma**	+	promote tumor growth	undefined	promote/inhibit tumor growth	NGF	β-blocker drugs	[29,39,53,82,83,86,100,146]
**PDAC**	+	inhibit tumor growth	promote tumor growth	promote tumor growth	BDNF; GDNF; NGF	NGF inhibitors	[35,37,38,61,108,109,112,120,121,122]
**CCA**	+	promote tumor growth	undefined	undefined	TGFβ	undefined	[62,108,110]
**HCC**	undefined	promote tumor growth	promote tumor growth	undefined	BDNF; NGF	β-blocker drugs	[131,137,145,147]

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
