# Peer review of "Roles of Peripheral Nerves in Tumor Initiation and Progression"

_ijms, 2025, doi:10.3390/ijms26157064_

Round 1

Reviewer 1 Report

Comments and Suggestions for Authors

This review summarizes the findings on the roles of peripheral nerves in different cancers.  The authors did a good job of providing the status of SCs in different cancer studies (categorized based on various organs). But like the authors mentioned, the findings are, in many cases, controversial. This makes it necessary for the authors to make some critical reviewing comments based on their understanding/opinion/experience in this review manuscript. Also, how SCs affect the cancer cells is still unclear after reading the manuscript. I would suggest that the authors provide some similar but more specific illustrations (or results from the literature) than what is currently shown in Figure 2 on how SCs affect each organ towards cancer for Section 4.

Author Response

Reply to the reviewer n. 1

Question n.1: “The authors did a good job of providing the status of SCs in different cancer studies (categorized based on various organs). But like the authors mentioned, the findings are, in many cases, controversial. This makes it necessary for the authors to make some critical reviewing comments based on their understanding/opinion/experience in this review manuscript”

As requested, we have expanded the Conclusions section by adding some critical comments.

Question n.2: I would suggest that the authors provide some similar but more specific illustrations (or results from the literature) than what is currently shown in Figure 2 on how SCs affect each organ towards cancer for Section 4.

As requested, an additional figure has been added (Figure 3) illustrating molecular pathways and secreted factors of activated SCs that potentially contribute to both nerve regeneration and tumor progression in PDAC, CCA and melanoma.

Reviewer 2 Report

Comments and Suggestions for Authors

A very good narrative review discussing the role of peripheral nerve component in the initiation and propagation of malignancies

Title- simple as a suggestion maybe title would be better focusing on the main concept of this review- that is tumor occurrence and progression. For sure peripheral nerves do have a role in regeneration however both discussion and text would become too complex and extended focusing on both. Authors do, indeed, describe very well the role of Schwan cells in peripheral nerve regeneration as a modality of argumentation which is very well placed, however the manuscript does not (and could not, of course) focus on peripheral nerve role in regenerative processes overall.

 Similarly, maybe replacing “cancer” with “tumors” would read somehow more appropriate

In the subchapter 4. (Schwan cells and cancer, actually the authors rather focus on molecular pathways active in these cells , maybe this would be worth mentioning in t he title. A short paragraph or table (or even graphical representation if possible) displaying side by side active pathway described in peripheral nerve regeneration Schwann cells and those involved in tumor progression would be of use.

Excellent description of Schwan cell embryo origin. Are there similar data for human development?

The manuscript goes on presenting several examples of malignancies and currently known influence of peripheral nerve proliferation.

Is there any common trait persisting across these various examples ? Is there a known gene expression profile or proteomics data available to compare Schwan cells across different malignancies?

Is there a potential systemic level involvement of PNS proliferation during tumor progression (in terms of pain sensitization, metabolism?)

Authors should briefly discuss the role of Schwann cells in acting as immune modulators as APC, cytokine production, immune modulation and if there are available data regarding their role in tumor related immune suppression  (Zhang et all 2020)

What do authors consider could be the practical outcomes of identifying PNS related  tumor proliferation  Maybe the two phrase paragraph in the conclusion chapter could be expanded and enriched mentioning how this could help in early diagnostic , drug sensitivity or in identification of perhaps novel therapeutic targets.

Author Response

Reply to the reviewer n. 2

Question n.1: Title- simple as a suggestion maybe title would be better focusing on the main concept of this review- that is tumor occurrence and progression.

The title has been changed as requested. The previous title was the following: “Roles of peripheral nerves in cancers: from regeneration to cancer initiation and progression”. The proposed new title is the following: “Roles of peripheral nerves in tumor initiation and progression.”

Question n.2: replacing “cancer” with “tumors” would read somehow more appropriate

The change has been made throughout the text whenever suitable.

Question n.3: graphical representation if possible displaying side by side active pathway described in peripheral nerve regeneration Schwann cells and those involved in tumor progression would be of use.

As requested, an additional Figure 3 has been added in the Conclusions Section. This novel figure illustrates molecular pathways activated in SCs, as well as factors secreted by SCs involved both in nerve regeneration and in tumor progression.

Question n.4 Excellent description of Schwan cell embryo origin. Are there similar data for human development?

The embryonic origin of Schwann cells (SCs), which are glial cells found in the peripheral nervous system, is well studied in mouse models. Data on human Schwann cell development is more limited. SCs have been derived in humans from embryonic stem cells (ESCs); induced pluripotent stem cells (iPSCs) display activation of a neural crest pathway and express specific markers (Sox10, p75NTR, S100, GFAP, MBP, MPZ, and Krox20).

There is a high degree of conservation in the developmental pathways between murine and human SCs, particularly regarding neural crest origin and early lineage specification (Ma MS, Boddeke E, Copray S. Pluripotent stem cells for Schwann cell engineering Stem Cell Rev Rep. 2015 Apr;11(2):205-18. doi: 10.1007/s12015-014-9577-1. PMID: 25433863).

Thus, the sentence “Although data on the embryonic origin of SCs in humans are more limited, they nevertheless suggest a high degree of concordance with those obtained in mice.” has been added to page 3, lines 83-85.

Question n.5 Is there any common trait persisting across these various examples? Is there a known gene expression profile or proteomics data available to compare Schwann cells across different malignancies?

We thank the reviewer for this comment. The following sentence is now present on pages 12-13 with an additional reference (157), lines 541-548:

“While the gene and protein profiling of SCs derived from malignant peripheral nerve sheath tumors as compared to normal SCs has been reported [157], the genomic and proteomic analysis of these glial cells within the context of different tumors remains to be investigated. It would be of interest to determine whether SCs occurring in solid tumors share common features or are instead related to tumor type and stage. In this context, as already mentioned, SCs appear to have an analgesic effect in the early stages of tumor growth, while increasing nociceptive activity in the later stages. The specific characteristics of SCs present in the early/late stages of tumor progression remain to be determined.”

More in detail to comply with the reviewer’s request:

Genomic and proteomic analysis of SCs within the context of other tissues cancers is still poorly investigated. However, some studies address the gene and proteins profiling in SCs derived from malignant peripheral nerve sheath tumors (MPNST) as compared to normal SCs. Miller and colleagues carried out a gene expression profiling to compare cell lines from MPNST to normal human SCs samples.  A 159 genes signature was identified containing genes downregulated such as SOX10, CNP, PMP22, NGFR, and genes upregulated such as SOX9 e TWIST1 (Miller 2006). Further, the human primary SCs secretome was investigated in a pancreatic cancer (PC) setup. Proliferation and/or invasion induced by SCs was found to depend on a number of proteins secreted by SC, namely matrix metalloproteinase-2, cathepsin D, plasminogen activator inhibitor-1, galectin-1, proteoglycan biglycan, galectin-3 binding protein, TIMP2 (Ferdoushi 2020). An additional study investigated the secretome derived from a MPNST-derived SCs, leading to the validated identification of 13 proteins involved in nerve tumorigenesis, namely Decorin, Lumican, Periostin, Gal-3BP, MMP-2, Cathepsin D, PAI-1, SPARC, Biglycan, GAL-3, TIMP-1, TIMP-2, Gal-1 (Ferdoushi et al. 2022). Finally, 1327 genes were found to be differentially expressed in immortalized SCs derived from MPNST, leading to the identification of Notch and Sonic Hedgehog (SHH) signaling as possible therapeutic targets (Bhunia et al. 2024).

Miller et al. Cancer Research 2006 Large-scale molecular comparison of human Schwann cells to malignant peripheral nerve sheath tumor cell lines and tissues PMID: 16510576 DOI: 10.1158/0008-5472.

Ferdoushi A et al.  Schwann Cell Stimulation of Pancreatic Cancer Cells: A Proteomic Analysis. Front Oncol 2020; PMID: 32984024; DOI: 10.3389/fonc.2020.01601

Ferdoushi A et al.  Secretome analysis of human schwann cells derived from malignant peripheral nerve sheath tumor. Proteomics 2022 Jan;22(1-2):e2100063. PMID: 34648240; doi: 10.1002/pmic.202100063

Bhunia et al.  Multiomic analyses reveal new targets of polycomb repressor complex 2 in Schwann lineage cells and malignant peripheral nerve sheath tumors. Neurooncol Adv. 2024 Nov 9;6(1). PMID: 39620202; doi: 10.1093/noajnl/vdae188

Question n.6 Is there a potential systemic level involvement of PNS proliferation during tumor progression in terms of pain sensitization, metabolism?

The relationship between increased tumor innervation and systemic effects remains highly uncertain. We have added a comment on this point in the Conclusions (page 13, lines 545-548), referring to what was already reported on page 8 in the sub-chapter “Neuropathic changes”.

Question n.7 Authors should briefly discuss the role of Schwann cells in acting as immune modulators as APC, cytokine production, immune modulation and if there are available data regarding their role in tumor related immune suppression (Zhang et all 2020)

In the Conclusions (page 13, lines 549-557) we have added the following sentences and reference 158:

“To note, recent research has convincingly demonstrated the SCs active involvement in modulating immune responses in a variety of pathological conditions, including neuropathic pain as well as cancer. Chemokines and cytokines secreted by SCs attract and influence immune cells. Specifically, SCs produce signals that attract M2-type macrophages (which suppress immune activation) and recruit MDSCs, thereby enhancing their capacity to inhibit T-cell proliferation. SCs can also directly suppress effector T cells, which in turn can push them towards a regulatory or exhausted state [158]. The immunosuppressive activity of SCs certainly constitutes a potential new therapeutic target to be explored in future studies.”

Question n.8 What do authors consider could be the practical outcomes of identifying PNS related  tumor proliferation.  Maybe the two phrase paragraph in the conclusion chapter could be expanded and enriched mentioning how this could help in early diagnostic, drug sensitivity or in identification of perhaps novel therapeutic targets.

As requested, we have expanded the Conclusions section by adding critical comments on this topic.

Reviewer 3 Report

Comments and Suggestions for Authors

Giampietri et al. provide an overview of the role of the peripheral nervous system and, in particular, Schwann cells (SCs) in developing various types of cancerous tumors. After a brief review of the embryonic formation process of SCs and their role in regeneration, the authors note that SC participation in the formation of malignant tumors exhibits features similar to regenerative activity.

The authors review the phenomenological aspects and the receptor and signaling mechanisms that mediate SC involvement in melanoma, pancreatic, and liver-derived tumor development (e.g., pancreatic adenocarcinoma, cholangiocarcinoma, and hepatocellular carcinoma). SC involvement in carcinogenesis is primarily associated with the factors they secrete, attraction of immune cells, polarization of those cells into an immunosuppressive phenotype, enhancement of the procarcinogenic activity of fibroblasts, extracellular matrix remodeling, and direct effects on cancer cell activity. The authors summarize the information in a table that provides an overview of possible ways to influence the development of neurogenic tumors.

Overall, the work is impressive, and the reviewer has no significant comments.

Small remarks:

  1. Question for the authors: Are SCs a trigger or a promoting factor for cancer development? Could there be limitations in the use of drugs that modulate SC activity in cancer treatment?
  2. Line 155: propranol
  3. Please give complete names of some abbreviated factors (Line 224-CGPR; Line 290-291 TrkA, etc).
  4. Table 1: Therapeutic targets=drugs (or potential drugs).

Author Response

Reply to the reviewer n. 3

Are SCs a trigger or a promoting factor for cancer development? Could there be limitations in the use of drugs that modulate SC activity in cancer treatment?

Schwann cells are traditionally known for their role in nerve repair, but they represent a promoting factor in cancer development. They can promote cancer cell migration, invasion, and even contribute to cancer-related pain. This happens via various mechanisms, e.g. the secretion of growth factors and chemokines, and by facilitating perineural invasion (PNI). Inhibiting Schwann cell function to block its pro-tumoral role is plausible, but there are challenges. Schwann cells play a crucial role in normal nerve function, so inhibiting them indiscriminately could cause neuropathy. However, by specifically blocking pathological activation (such as the transition to a pro-invasive 'reparative' phenotype), one could interfere with tumor-nerve cooperation. Some known molecular targets are signalling pathways that are active in both tumor cells and Schwann cells, such as TGF-β, c-Jun, NF-κB and CXCL12/CXCR4.

Minor points

  • propranolol spelling has been corrected at line 154
  • Complete names have been indicated for Calcitonin Gene-Related Peptide (CGPR) and Tyrosine kinase receptors (Trk).
  • In Table 1: Therapeutic targets have been changed in Potential drugs as suggested.
